# A New Circuit Design of AC/DC Converter for T8 LED Tube

**Sunghwan Kim** [1] and **Haiyoung Jung** [2,*]

1   Electrical Engineering Department, Inha University, 100 Inharo, Nam-gu Incheon 22212, Korea; saint1119@naver.com

2   Fire and Disaster Prevention Department, Semyung University, 65 Semyung-ro, Jecheon-si, Chungcheongbuk-do 27136, Korea

*   Correspondence: hyjung@semyung.ac.kr; Tel.: +82-43-649-1695

**Abstract:** This study is about an improved high-quality light-emitting diode (LED) converter for a T8 LED tube. The converter is separated into the AC driving circuit and DC driving circuit. Also, the LED tube was applied with an output ripple eliminator for the optical performance. The AC driving circuit and DC driving circuit are assembled at the end of the LED tube in a G13 base and a G13 base dummy, respectively, and the output ripple eliminator is located on an LED PCB. The proposed LED converter is founded on a SSBB (single-stage buck-boost) converter topology and was designed for 10 W operation for a 600 mm T8 LED tube. The light waveform of the LED tube was measured by a photosensor. The waveform had almost no ripple and was the same as a straight line. The average calculated percent flicker of the proposed LED converter was an average of 1.9% at 100 and 240 VAC, 50 and 60 Hz. The proposed converter has lower power efficiency than a conventional converter by 2.7% at 100–240 VAC, but it still has high power efficiency (>87%). The measurement results represent that the LED output current regulation is below 0.92% at 100–240 VAC and the converter obtains the power factor more than 0.84 and the total harmonic distortion is less than 14.3%. All of the current harmonics reach the IEC 61000-3-2 Class D standards for high-quality LED converters.

**Keywords:** high circuit efficiency; photosensor; power factor correction; SSBB converter topology; ripple

## 1. Introduction

LEDs have a lot of characteristics such as high luminous efficacy, energy saving properties and long lifetimes. These advantages have allowed LED lighting to replace other types of lighting quickly in the marketplace. Furthermore, as regulations for the use of traditional lighting such as incandescent lamps and florescent lamps become increasingly stronger, the market share of LED lighting is expected to increase rapidly [1–3].

An LED lighting product generally consists of various mechanical parts, an LED module and an LED converter. An LED converter should provide stable and accurate current to the LED module to achieve good optical performance, since the luminance variation of LEDs depends on the variation of the current supplied to LEDs. In addition, the LED converter requires high power efficiency, high power factor (PF), low total harmonic distortion (THD), low total cost, and low light flicker [4,5].

The light quality of LED lighting is mainly determined by the driving method and the key components of the LED converter. Due to the low circuit complexity and low cost, a single-stage power factor correction (PFC) driving method is usually used in many types of low-power LED lightings, including LED tubes. However, the output current inevitably has a double line frequency ripple and light flicker is generated by variation of the luminance. Various studies show that low-frequency light flicker caused by large output current ripple can adversely affect human health and cause headaches, visual fatigue and epileptic attacks [6–9].

Figure 1 shows a block diagram of a typical single-stage buck-boost LED converter, which is composed of many function blocks. An AC voltage with 50/60 Hz is supplied to a

bridge rectifier through a line filter, and a full-wave rectified sinusoidal voltage is supplied to the DC link capacitor $C_{LINK}$. Switching is conducted for power factor correction to obtain high power factor, and smoothing is performed to reduce the output ripple and control the output current [5].

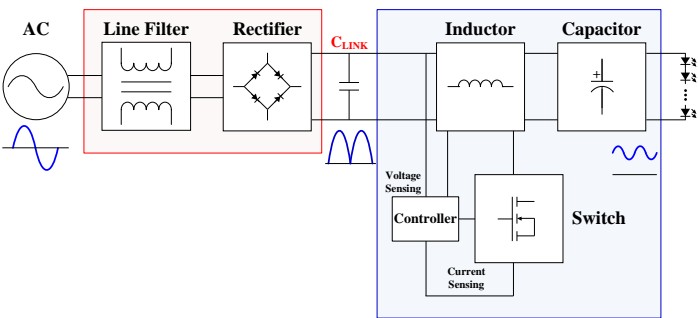

**Figure 1.** The block diagram of typical single-stage buck-boost light-emitting diode (LED) converter.

The smoothing to remove the ripple is not perfect, so the single-stage PFC driving method always makes a low-frequency output ripple, and output current varies between a maximum peak and minimum peak [10]. If the light has cyclic variation in the amplitude, percent flicker is very useful to represent the level of light flicker, which is defined as [11]:

$$\text{Percent Flicker }(\%) = \frac{A - B}{A + B} \times 100 \tag{1}$$

where A and B are the maximum and minimum luminance, respectively. The Equation (1) shows that the low percent flicker represents good performance. Several methods can be considered to improve the light flicker caused by the double line frequency output ripple [12–15]. One of the obvious ways is to increase the link capacitor $C_{DC}$; however, regulatory requirements for PF and THD performance may not be met. Therefore, this method is usually suitable for non-regulated low power LED lighting. Another method is to increase the output capacitance for the smoothing operation. However, a large electrolytic capacitor is needed to remove the output ripple completely, which increases the system size and cost. Therefore, this is not suitable for small lighting applications with small circuit space. Various converter topologies have been proposed to remove current ripple [16–18]. In [16], a flicker-free LED converter composed of PFC flyback converter and a bidirectional buck/boost converter to reduce the current ripple was proposed to reduce the current ripple. The output current waveform is almost flat, while high power efficiency is maintained [16,17].

However, due to the high circuit complexity, it cannot be applied to applications such as T8 LED tubes. A multiplexing ripple cancellation was proposed by adding a ripple cancellation unit to achieve flicker-free operation in [18]. Although this solution is very competitive, it is still too large to mount inside a G13 base with a diameter of 28 mm for low-power LED tubes [19]. In this paper, an LED converter with an output ripple eliminator is proposed to remove the double line frequency flicker of a T8 LED tube, while providing high power efficiency, high PF, low THD and precise output current regulation. Due to the addition of the output ripple eliminator, the power efficiency has been slightly reduced by an average of 2.7%. Nevertheless, the power efficiency is still more than 87% and light flicker is almost removed. The proposed LED converter is separated into AC and DC driving circuit and designed to mount in a G13 base and G13 base dummy. Because of the size limitations, we propose a practical method for product manufacturing by locating the output ripple eliminator on the LED module. In Section 2, the theory of the output ripple eliminator operation is explained. In Section 3, design specifications of the proposed LED converter are represented in detail. In Section 4, a 10 W prototype is introduced and experimental results are discussed. Finally, conclusions are given in Section 5.

## 2. The Proposed LED Converter for a T8 LED Tube

### 2.1. Power Stage

Figure 2a shows the overall stages of the proposed LED converter, which consists of a power stage for constant current control and a ripple remove stage to reduce the output ripple. The power stage is physically divided into two driving circuit that are electrically linked though the LED module [5]. Figure 2b shows a simplified circuit diagram of the proposed LED converter with separated driving circuit for a T8 LED tube.

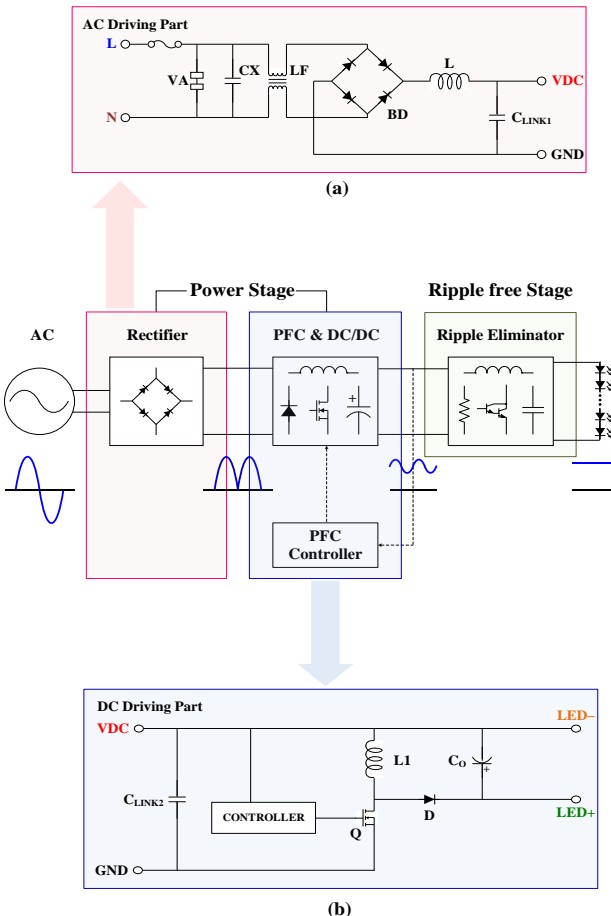

**Figure 2.** (**a**) Overall stages of the proposed LED converter; (**b**) simplified circuit diagram of AC and DC driving circuit.

The proposed LED converter employs a single-stage buck-boost converter topology and consists of an AC driving circuit for rectification and a DC driving circuit for constant current control. The AC driving circuit consists of various filter components (CX, LF) for high frequency suppression, a full bridge rectifier (BD) and a DC link capacitor ($C_{LINK1}$). The DC driving circuit includes another DC link capacitor ($C_{LINK2}$), a power switch for energy transfer (S), an inductor (L), a diode (D) and an output capacitor ($C_O$) [5,20]. The full bridge rectifier generates a positive full wave sinusoidal voltage from the AC line and supplies it to the DC link capacitor in the AC driving circuit. The inductor is magnetized and stores rectified input energy from the AC driving circuit when the switch S turns on.

During the turn-off time, the inductor is demagnetized, and stored energy is transferred to the output capacitor. In the single-stage driving method, the input voltage can be defined as:

$$v_{IN}(t) = V_{in} \cdot sin(2\pi f_L t) \tag{2}$$

where $V_{in}$ is the amplitude of the input voltage and $f_L$ is the line frequency of the input voltage. Since the controller is operated for power factor correction, the input current waveform follows the input voltage waveform. Therefore, the input current is defined as:

$$i_{IN}(t) = I_{in} \cdot sin(2\pi f_L t) \tag{3}$$

where $I_{in}$ is the peak value of the input current. From Equations (1) and (2), the instantaneous output power can be expressed as [5,21]:

$$p_{OUT}(t) = \eta_{eff} \cdot p_{IN}(t) = \eta_{eff} \cdot V_{in} \cdot I_{in} \cdot sin^2(2\pi f_L t) = P_o \cdot [1 - cos(4\pi f_L t)] \tag{4}$$

where $\eta_{eff}$ is the expected power efficiency and $\eta_{eff} \cdot V_{in} \cdot I_{in}/2$ is substituted with the amplitude of the output power $P_o$. When $V_o$ is the output voltage, the instantaneous output current can be expressed as:

$$i_{OUT}(t) = I_o \cdot [1 - cos(4\pi f_L t)] \tag{5}$$

where $I_o$ is the amplitude of the output current. The equation (5) shows that the single-stage power factor correction driving method always contains double line frequency output ripple. If the output capacitance does not sufficiently smooth the output ripple, the high and low amplitude current flowing through the LEDs generates light flicker [5,21–23].

*2.2. Ripple Free Stage*

Figure 3 shows the output ripple eliminator of the proposed LED converter for alleviating light flicker. An output ripple eliminator is a kind of common collector amplifier using an NPN Darlington configuration, which is one of the basic amplifier topologies, as a voltage follower. It consists of $R_E$, $C_E$, and a Darlington transistor $Q$ [24,25].

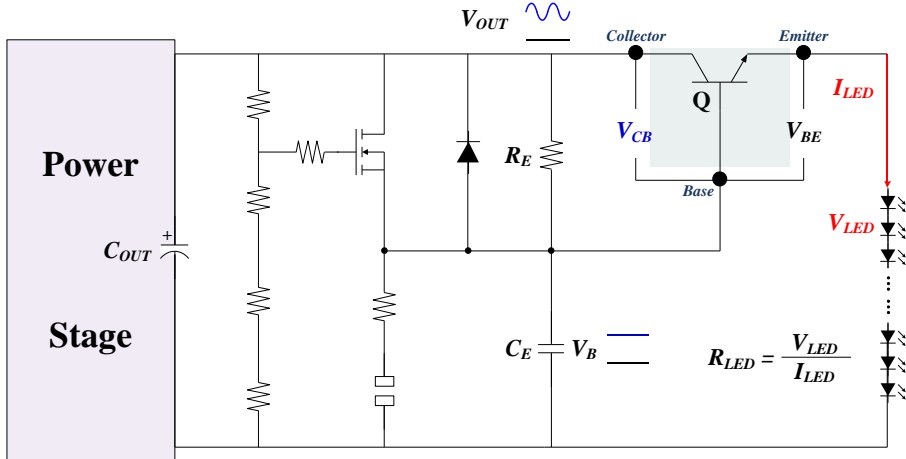

**Figure 3.** The output ripple eliminator.

Since the bias voltage of the base is immediately transferred to the LED anode, the impedance should be carefully chosen to find the optimized LED operation point in terms of power losses and light flicker elimination.

Figure 4 shows conceptual waveforms of the proposed output ripple eliminator. To achieve a high power factor of the proposed LED converter, the output waveform inevitably contains a large ripple with double line frequency. This double frequency ripple is passed through the power stage and transferred to the ripple removal stage. Since the output voltage $V_o$ is constant with output ripple, it can be expressed as:

$$V_O(t) = V_{O\_dc} + V_{O\_ac}(t) \tag{6}$$

where $V_{O\_dc}$ and $V_{O\_ac}(t)$ are the DC component and AC component with double line frequency ripple, respectively. Therefore, the base bias voltage of the Darlington pair has the same frequency ripple in steady state and the instantaneous value of bias voltage can be expressed as:

$$V_B(t) = V_{B\_dc} + V_{B\_ac}(t) = (V_{O\_dc} - V_{CB\_dc}) + [V_{O\_ac}(t) - V_{CB\_ac}(t)] \tag{7}$$

where $V_{CB\_dc}$ and $V_{CB\_ac}(t)$ are the DC component and AC component of the collector-base forward voltage, respectively. Considering the base-emitter forward voltage in the Darlington input stage, $V_{LED}$ applied across LEDs can be obtains as:

$$V_{LED}(t) = (V_{B\_dc} - V_{BE\_dc}) + [V_{B\_ac}(t) - V_{BE\_ac}(t)] \tag{8}$$

where $V_{BE\_dc}$ is the DC value, and $V_{BE\_ac}(t)$ is the AC value. Using Kirchhoff's voltage law, the AC component of bias voltage $V_{B\_ac}(t)$ is determined from $X_E$, $Z_E$ and $V_{O\_ac}(t)$,

$$V_{B\_ac}(t) = \frac{X_E}{Z_E} \cdot V_{O\_ac}(t) \tag{9}$$

where $Z_E = \sqrt{R_E{}^2 + X_E{}^2}$ is the total impedance, and $X_E = 1/(4\pi f_L \cdot C_E)$ is the impedance of $C_E$. If the base-emitter forward voltage $V_{BE}$ components are negligible, from Equations (8) and (9), the LED voltage can be derived as:

$$V_{LED}(t) = (V_{O\_dc} - V_{CB\_dc}) + \frac{X_E}{\sqrt{R_E{}^2 + X_E{}^2}} \cdot V_{O\_ac}(t) \tag{10}$$

Equation (10) shows that the output ripple eliminator can provide a DC value to the LEDs by adjusting the AC value to nearly zero, which means light flicker completely disappears from the LED tube.

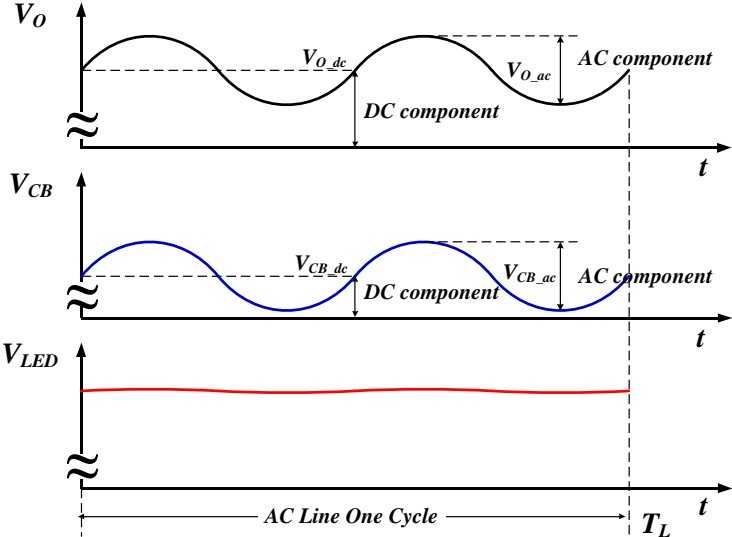

**Figure 4.** Conceptual waveforms of the output ripple eliminator.

## 3. Design Parameters

Key parameters and components were selected for the converter. Table 1 shows the specifications of the proposed LED converter. It operates at 100–240 VAC and is designed to cover ±10% input variation. The nominal power consumption is 10 W with ±10% tolerance. The target variation of the output voltage and current are 106 V ± 4% and 87 mA ± 6%, respectively.

**Table 1.** Specifications of the proposed LED converter.

| 10 W LED Converter | | | | |
|---|---|---|---|---|
| **Item** | | **MIN** | **TYP** | **MAX** |
| Input Voltage (VAC) | 100 | 90 | 100 | - |
| | 240 | - | 240 | 264 |
| Input Current (mA) | | 34.1 (@90 VAC) | - | 122.2 (@264 VAC) |
| Input Frequency (Hz) | | 50 | - | 60 |
| Input Power (W) | | 9.0 | 10.0 | 11.0 |
| Output Voltage (V) | | 102 | 106 | 112 |
| Output Current (mA) | | 81.8 | 87.0 | 92.2 |

### 3.1. Determining the Inductance: L

In a discontinuous current mode(DCM) single-stage buck-boost converter, the switching cycle time ($T_{SW} = 1/f_{SW}$) can be expressed as [5]:

$$T_{SW} = T_{ON\_S} + T_{ON\_D} + T_{IDLE} \tag{11}$$

where $T_{ON\_S}$ is the switch turn-on time, $T_{ON\_D}$ is the diode on-time and $T_{IDLE}$ is the time, in which no current flows in the inductor. Since $T_{IDLE}$ cannot be zero to ensure DCM operation under all conditions, $D_S$ and $D_D$ should satisfy $D_S + D_D < 1$. Therefore, from the volt-second balance law, the inequality for $D_S$ becomes:

$$D_S < \frac{V_O}{V_{IN} + V_O} \tag{12}$$

where $D_S$ is the duty ratio of the switch turn-on time ($= T_{SW}/T_{ON\_S}$) and $D_D$ is the duty ratio of the diode turn-on time ($= T_{SW}/T_{ON\_D}$). The input current is induced as $D_S^2 \cdot T_{sw} \cdot V_{IN}/2L$ from the buck-boost relationship, which is substituted into the power conversion relationship ($V_O \cdot I_O = \eta_{eff} \cdot V_{IN} \cdot I_{IN}$) and rearranged for the inductance L:

$$L < \frac{\eta_{eff}}{2f_{sw}} \cdot \frac{V_O}{I_O} \cdot \left( \frac{V_{IN}}{V_{IN} + V_O} \right)^2 \tag{13}$$

where $f_{SW}$ is the switching frequency. From Equation (13), the minimum inductance can be determined at $V_{IN\_MIN}$, $V_{O\_MAX}$ and $f_{SW\_MAX} = 70\ kHz$.

$$L < \frac{0.9 \times 112}{2 \times 70\ k \times 92.2\ m} \cdot \left( \frac{90}{90 + 112} \right)^2 = 1.55\ mH \tag{14}$$

Considering the size of inductor and margin, an inductance of 1.38 mH should be chosen and manufactured with an EE1616 core.

### 3.2. Determining the Output Capacitance: $C_O$

A small output ripple requires a large output capacitance; however, the higher output capacitance, the bigger the capacitor size. Therefore, a proper percent flicker specification without an output ripple eliminator should be considered to determine the output capacitance while considering voltage stress and size. The output capacitance $C_O$ has the following relationship [5,24]:

$$C_O \approx \frac{I_{C\_PP}}{4\pi f_L \cdot V_{C\_PP}} \approx \frac{2 \cdot I_{LED}}{4\pi f_L \cdot N_S \cdot \Delta V_{LED\_Device}} \tag{15}$$

where $I_{C\_PP}$ is the peak-to-peak current of the output capacitor, which is approximately twice the average LED current $I_{LED}$. Here, $\Delta V_{LED\_Device}$ is the one LED peak-to-peak, which varies with the LED current and $N_S$ is the number of LEDs in series. Referring to Equation (15), the minimum output capacitance can be estimated with $I_{LED\_MAX}$ of 92.2 mA and $f_{L\_MIN}$ of 50 Hz. In addition, to keep the light flicker around 10% without output ripple eliminator, $\Delta V_{LED\_Device}$ of 95 mV should be used according to V-I characteristics [26].

$$C_O > \frac{2 \times 92.2\ m}{4 \times 3.14 \times 50 \times 35 \times 95\ m} = 88.3\ \mu F \tag{16}$$

For practical design for a T8 LED tube, $C_O$ should be chosen as 160 V/100 μF with dimensions of $12 \times 25$ mm$^2$ (D × H).

### 3.3. Determining the Power Device: S, D

Both the voltage and current stress should be calculated to select an appropriate device. Since the voltage stress of switch is $V_{IN} + V_O$ in a buck-boost converter topology, the maximum voltage stress is determined at $V_{IN\_MAX}$ and $V_{O\_MAX}$. Therefore, the voltage stress of the switch can be calculated as:

$$V_{S\_MAX} = \sqrt{2} \cdot V_{IN\_MAX} + V_{O\_MAX} = \sqrt{2} \cdot 264 + 112 = 485\ V \tag{17}$$

From Table 1, when an input voltage of 90 VAC is provided to an 11 W system, the maximum RMS current stress of the switch when considering the expected power efficiency is:

$$I_{S\_RMS\_MAX} = \frac{P_{IN\_MAX}}{V_{IN\_MIN} \cdot \eta_{eff}} = \frac{11}{90 \times 0.9} = 136\ mA \tag{18}$$

The voltage stress of the diode is the same as $V_{S\_MAX}$ and the maximum average current of the diode is:

$$I_{D\_AVG\_MAX} = \frac{2 \cdot I_{O\_MAX}}{D_D} = \frac{2 \times 92.2\ m}{0.44} = 419\ mA \tag{19}$$

Therefore, an 800 V/2.5 A MOSFET switch (STP3NK80) and 1000 V/1 A diode (US1M) were selected to consider the component stress and margin with heat dissipation.

### 3.4. Determining the Output Ripple Eliminator: $R_E$, $C_E$, Q

The voltage and current relation is almost linear in driving region of LED IV curve [26], so the dynamic resistance of the LED device can be defined as:

$$R_{LED\_Device} = \frac{V_{LED\_Device}}{I_{LED\_Device}} \approx 3.23\ \Omega \tag{20}$$

To achieve a percent flicker under 1%, the current variation $\Delta I_{LED\_Device}$ of the LED device should be within 1.4 mA according to the luminance-current characteristics [26]. Consequently, the allowed LED device voltage $\Delta V_{LED\_Device}$ is 3.23 × 1.4 m = 4.5 mV. Therefore, when the peak-to-peak output voltage $\Delta V_{O\_ac}(t)$ is 3.33 V, the bias voltage $V_{B\_ac}(t)$ should be 157.5 mV in a series of 35 LEDs. From Equation (9), $X_E/Z_E$ is obtained as:

$$\frac{X_E}{\sqrt{R_E{}^2 + X_E{}^2}} = \frac{\Delta V_{B\_ac}}{\Delta V_{O\_ac}} = \frac{0.158}{3.33} = 0.047 \tag{21}$$

A small capacitor can be used to remove the low frequency ripple, and $C_E$ can be implemented with a 100 V/1 μF chip capacitor. Since the capacitor impedance $X_E$ is 1.59 kΩ at 60 Hz, $R_E$ should be:

$$R_E = \sqrt{\left(\frac{X_E}{0.047}\right)^2 - X_E{}^2} \approx 33\ k\Omega \tag{22}$$

The output ripple eliminator greatly reduces the light flicker but at the same time, additional power loss occurs in Darlington transistor Q. The power loss of Q can be estimated as $(V_{LED\_PP}/2 + V_{BE}) \cdot I_{LED}$, where the $V_{LED\_PP}$ is peak-to-peak voltage across the LED array and $I_{LED}$ is the average current flowing to the LED array [24,27].

$$P_{Q\_MAX} = \left( \frac{35 \times 95 \ m}{2} + 1.2 \right) \times 92.2 \ m = 264 \ mW \tag{23}$$

Considering Equation (23) and heat dissipation, an MJF6388 Darlington transistor is selected to maintain stable operation.

## 4. Experimental Results

A 10 W prototype of the LED converter was designed to perform the experiments, as shown in Figure 5. The input voltage is 100–240 VAC and the line frequency is 50 Hz and 60 Hz. The output current is 87 mA, and output voltage is 106 V with 35 LEDs in series to satisfy the luminous uniformity. Table 2 shows the specifications of the key components.

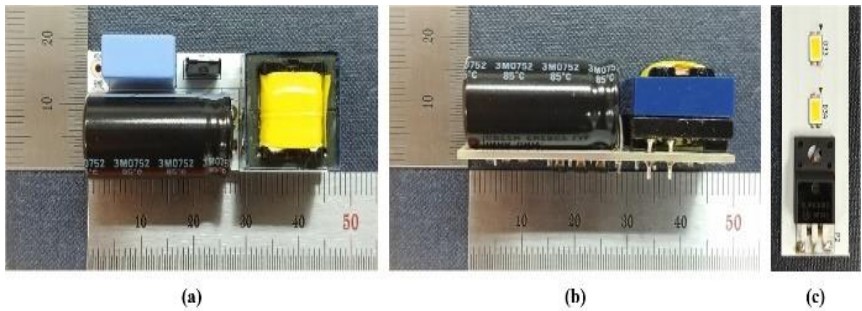

**Figure 5.** (**a**) AC driving circuit, (**b**) DC driving circuit, (**c**) LED module with the output ripple eliminator.

**Table 2.** Specifications of key components.

| | | |
|---|---|---|
| | Fuse | 250 V, 1 A |
| | Line Filter | TOROIDAL, 31 uH, 4 A |
| AC Driving Circuit | X-capacitor | 275 VAC, 100 nF |
| | Bridge Diode | VRRM = 1000 V, IO = 1 A |
| | Film Capacitor | 600 V, 100 nF |
| | PCB | CEM-3, 1T, 0.5 oz, Single Layer |
| | Film Capacitor | 630 V, 100 nF |
| | Inductor | EE1616, 1.38 mH |
| | Electrolytic Capacitor | 400 V, 22 uF, 105 °C |
| DC Driving Circuit | FET | VDSS = 800 V, ID = 2.5A, RDS(ON) < 4.5 Ω |
| | Diode | VRRM = 1000 V, IO = 1 A |
| | PCB | FR-4, 1T, 0.5 oz, Double Layer |
| | Darlington Transistor | NPN, VCE 100 V, IC = 10 A |

Figure 6 shows the input voltage and current waveform of the proposed LED converter. The input current waveform follows the input voltage, which means it operates properly for power factor correction. Since the PFC operation is not affected by the output ripple eliminator, there is no difference in the power factor between the reference converter and the proposed converter.

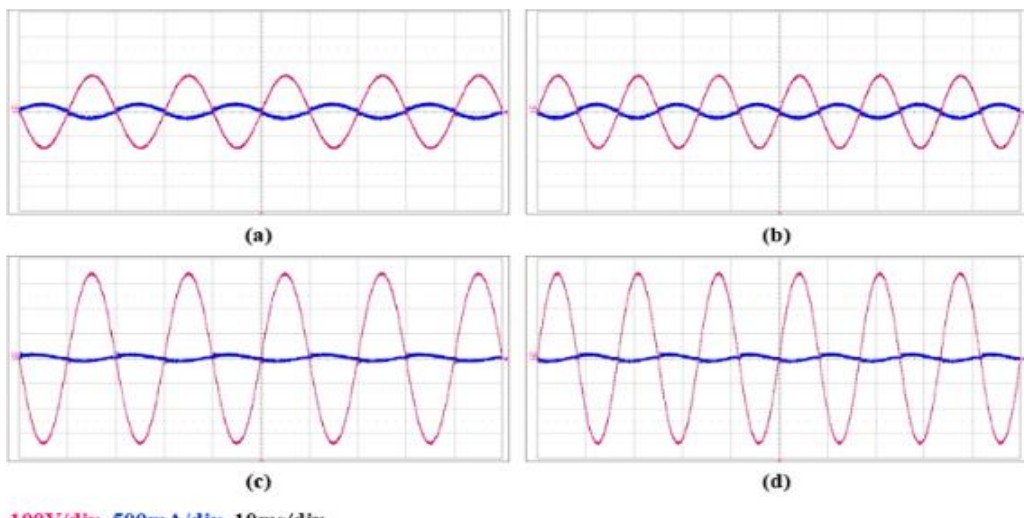

100V/div, 500mA/div, 10ms/div

**Figure 6.** Input voltage waveform (red) and input current waveform (blue) (**a**) at 100 VAC/60 Hz, (**b**) 100 VAC/50 Hz, (**c**) 240 VAC/50 Hz, (**d**) 240 VAC/50 Hz.

Figure 7 shows the measured waveform of the output voltage, output current, and light output at 240 VAC/50 Hz. The light output waveform measured with the photosensor amplifier contains a double line frequency ripple in the reference converter without an output ripple eliminator. However, with the proposed converter, the light flicker is completely removed and the result is almost a straight line.

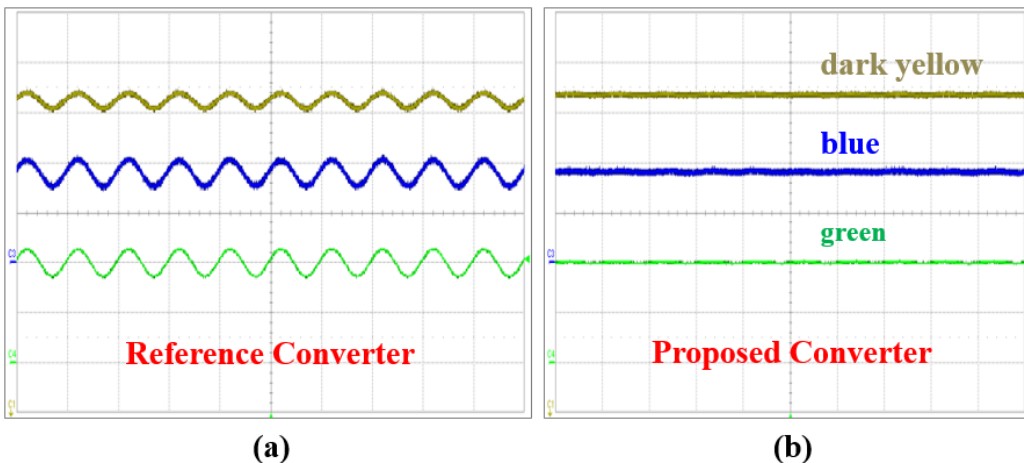

10V/div, 50mA/div, 200mV/div, 10ms/div

**Figure 7.** Waveform of the output voltage (dark yellow), output current (blue) and light output (green) of the (**a**) reference converter, (**b**) proposed converter.

Using Equation (1), the percent flicker can be calculated in each input condition. Table 3 shows the calculated percent flicker of the reference and the proposed converter which are 13.5% and 1.9%, respectively. When using the output ripple eliminator, the light flicker almost disappeared, and the percent flicker of the proposed converter with the output ripple eliminator was improved by 11.6% on average.

**Table 3.** Comparison of percent flicker.

| Input | | Reference | | | | Proposed | | | |
|---|---|---|---|---|---|---|---|---|---|
| | | Light Output (mV) | | | Percent Flicker (%) | Light Output (mV) | | | Percent Flicker (%) |
| VAC | Hz | MAX | AVG | MIN | | MAX | AVG | MIN | |
| 100 | 50 | 450 | 393 | 336 | 14.5 | 401 | 393 | 385 | 2.0 |
| | 60 | 443 | 394 | 345 | 12.5 | 402 | 395 | 388 | 1.8 |
| 240 | 50 | 451 | 394 | 337 | 14.5 | 401 | 393 | 385 | 2.0 |
| | 60 | 443 | 394 | 345 | 12.4 | 400 | 393 | 386 | 1.8 |

Figure 8 shows power efficiency and efficiency difference according to the input voltage measured at 50 Hz. There is no big variation according to the line frequency. The power efficiency of the proposed converter is 2.6–2.9% lower than that of the reference, which means the output ripple eliminator consumes less than 2.6–2.9% of power. Although the total power increases 2.7% on average, the power efficiency is still more than 87% in 100–240 VAC, and the light flicker can be completely removed.

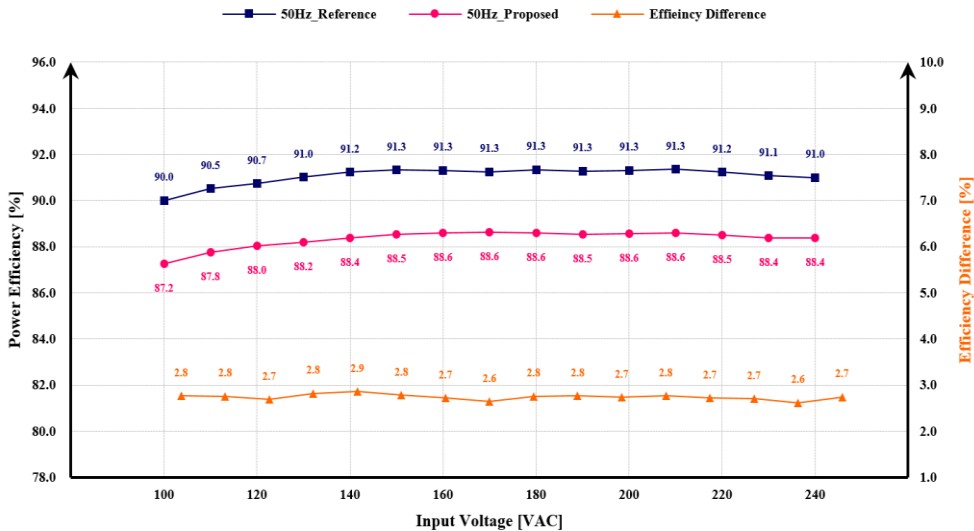

**Figure 8.** Power efficiency and efficiency difference according to the input voltage.

Figure 9 shows the measured output current of the reference and proposed LED converters, which was measured using an LED module with 35 LEDs in series. The measured data of the proposed LED converter is almost the same as that of reference, which shows that the output current regulation is hardly affected by the output ripple eliminator. The average output current and its standard deviation were 87.1 mA and 0.19 mA for the proposed LED converter at 60 Hz, respectively. The positive and negative maximum deviations from the average output current were measured as +0.3 mA (0.34%) and −0.2 mA (0.23%) at 60 Hz. The variation of the output current is less than 1%, which is great performance, since the power consumption tolerance of the LED tube is usually ±10%.

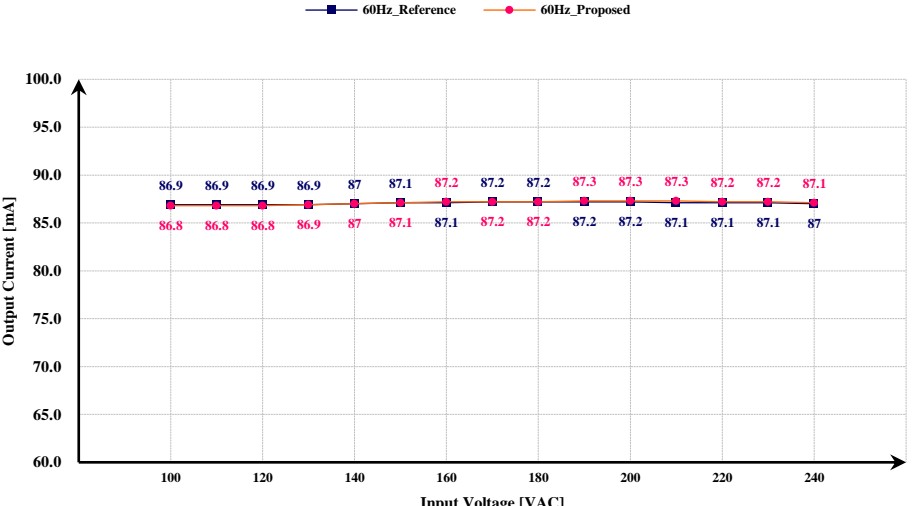

**Figure 9.** Output current according to the input voltage.

Figure 10 shows the measured power factor in 100–240 VAC at 60 Hz. There is no large difference between the reference and proposed LED converters, which means the variation of the power factor is not affected by the output ripple eliminator. Due to the influence of the input current harmonics, it slightly decreases as the input voltage increases. Nevertheless, the power factor is still higher than 0.84 under all input voltages.

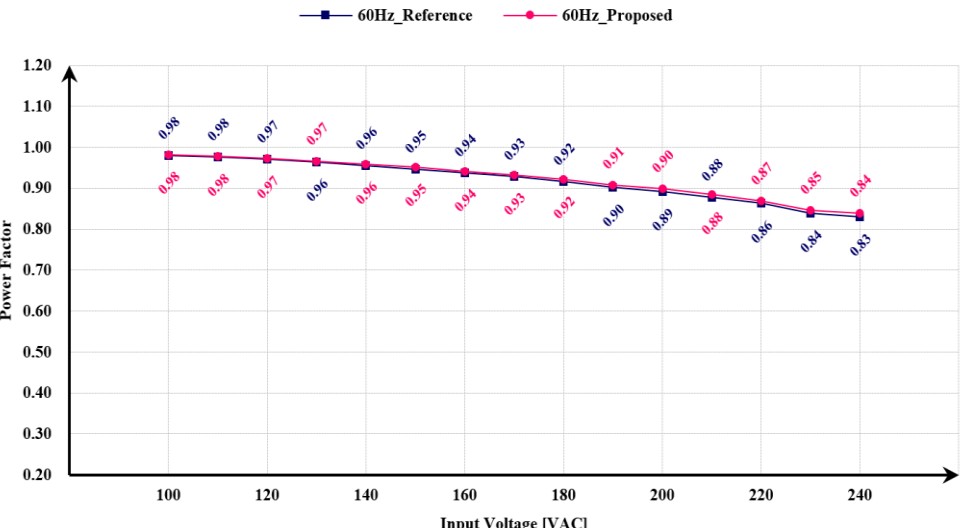

**Figure 10.** PF at 60 Hz.

Figure 11 shows the THD of the proposed LED converter at 50 Hz and 60 Hz, which is less than 14.3% under all input voltages. The values of THD at 100 VAC are 8.52% and 8.36% at 50 Hz and 60 Hz, respectively. At 240 VAC, the values are 11.44% at 50 Hz and 14.26% at 60 Hz, respectively. The measured results are slightly high at 60 Hz, but they are very suitable for a high-quality LED tube.

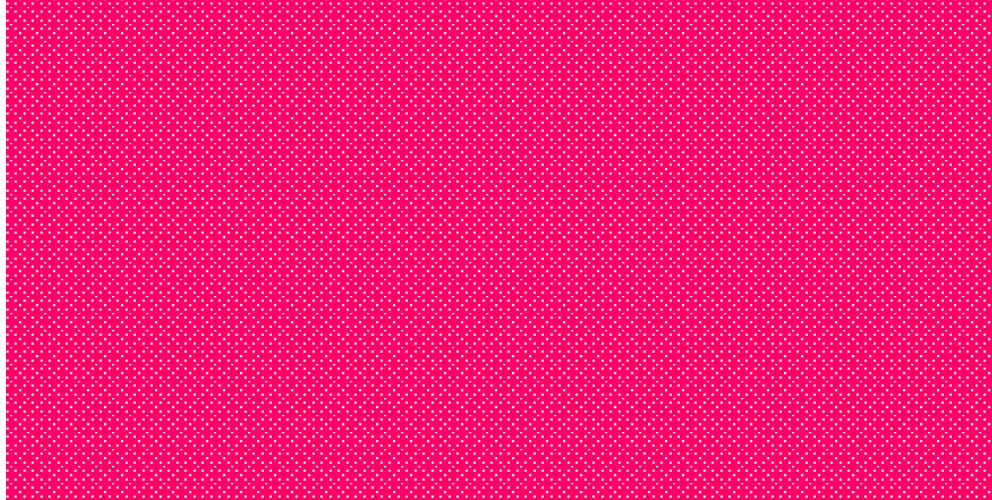

**Figure 11.** Total harmonic distortion (THD) according to the input voltage.

Figure 12 shows the measured results of the harmonic current at 100 VAC at 50 Hz and 60 Hz. The maximum permissible harmonic current from the IEC 61000-3-2 Class D standard for less than 25 W of lighting is presented with a gray bar [28,29]. The harmonic current is markedly lower than the IEC standards.

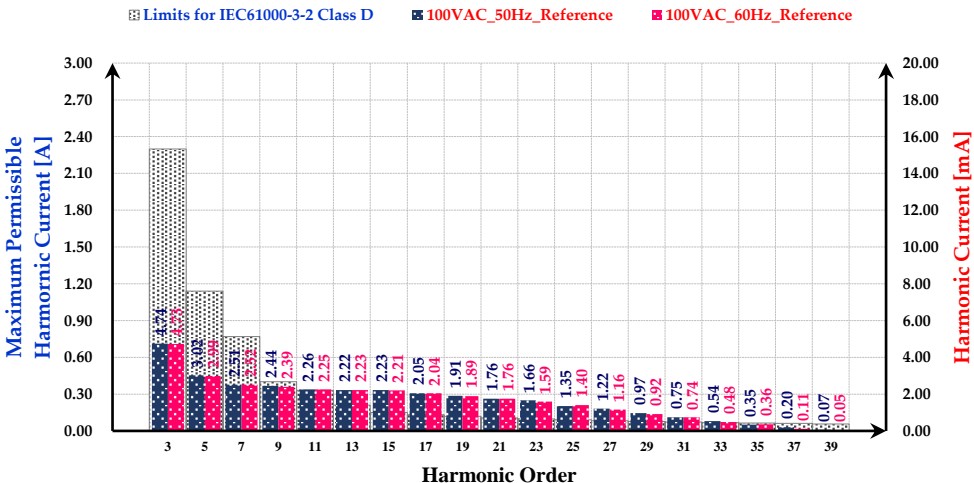

**Figure 12.** Harmonic current at 100 VAC 50/60 Hz.

## 5. Conclusions

In this study, an AC/DC LED converter has been proposed to remove the flicker of a T8 LED tube. The proposed LED converter uses an output ripple eliminator and a SSBB converter topology for the input voltage 100–240 VAC. A 10 W prototype was designed to verify the high performance. The calculated percent was 1.9% at all input conditions. The power efficiency is lower than that of a conventional converter by 2.7% at 100–240 VAC, but it is still high (>87% and even 89% at 220 VAC). The experimental results represented the LED output current regulation as less than 0.92% at 100–240 VAC and the LED converter has a high power factor (>0.84) and low total harmonic distortion (<14.3%). Moreover, the harmonic current of the LED converter reaches the IEC 61000-3-2 Class D standard at 100 VAC and 240 VAC input voltages for high-quality LED converters.

**Author Contributions:** Conceptualization, S.K. and H.J.; methodology S.K. and H.J.; software, S.K.; validation, S.K. and H.J.; formal analysis S.K. and H.J.; investigation, S.K. and H.J.; resources, S.K.; data curation, S.K.; writing—original draft preparation, S.K.; writing—review and editing, S.K. and H.J.; visualization, S.K. and H.J.; supervision, H.J. All authors have read and agreed to the published version of the manuscript.

**Funding:** This research received no external funding.

**Conflicts of Interest:** The authors declare no conflict of interest.

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
