# Peer review of "A New Circuit Design of AC/DC Converter for T8 LED Tube"

_applsci, doi:10.3390/app11010421_

Round 1
Reviewer 1 Report
Dear authors. Thank you for your contribution. The article is well-written, the figures are of good quality and it read smoothly.
The following points should be considered before publication:
- Use subscript for the notation of parameters. For instance on page 2 line 49 use CLINK rather than CLINK. Or VAC not VAC etc. Please skim through the entire manuscript and correct all similar notations.
- Line62: remove the indent before "where."
- Line 81: "Nevertheless, the power efficiency is still more than 87%"--> Why 87% is important? Is this a threshold value determined by standards? Please clarify.
- Figure 1 uses a transformer image for an inductor. It should be addressed.
- Line 305: What optical performance requirements are considered here?
- How realistic is it to use a 10-W prototype LED converter? How does it compare with commercial products?
- How much would the power consumption of the ripple eliminator increase when the experimental design is upscaled? It should be mentioned. Depending on the authors' response to the previous bullet point, it would be sound to extrapolate the losses of the ripple eliminator at the power levels of the commercial products. Would the efficiency still be above 87%?
Author Response
Point 1 : Use subscript for the notation of parameters. For instance, on page 2, line 49 use CLINK rather than CLINK. Or VAC not VAC etc. Please skim through the entire manuscript and correct all similar notations.
Response 1 : We acknowledge that use subscript for the notation of parameters. In the revised paper, CLINK reflected subscripts. However, VAC is a unit, other reference papers use this notation. And so, VAC notation is maintained in this revised paper.
Point 2 : Line 62: remove the indent before "where"
Response 2 : In the revised paper, remove the indent before "where" all over.
Point 3 : Line 81: "Nevertheless, the power efficiency is still more than 87%"--> Why 87% is important? Is this a threshold value determined by standards? Please clarify.
Response 3 : Ripple eliminator reduces power efficiency. The conventional LEDs showed a power efficiency of 87% with a flicker, while the proposed LEDs showed a power efficiency of 87% without a flicker. We think the above is included in the paper.
Point 4 : Figure 1 uses a transformer image for an inductor. It should be addressed.
Response 4 : In the revised paper, Figure 1 represents a transformer image for an inductor.
Point 5 : Line 305: What optical performance requirements are considered here?
Response 5 : In LED tubes, LEDs' arrangement is not continuous and arranged spaced. Due to these characteristics, Luminous uniformity becomes an important consideration. In this paper, optical performance requirements mean these optical properties like a luminous uniformity.
Point 6 : How realistic is it to use a 10-W prototype LED converter? How does it compare with commercial products?
Response 6 : Generally, the 600mm LED tube is present in most products 8 to 12W.Therefore, the 10-W prototype LED tube discussed in this paper is expected to be commercially viable as well.
Point 7 : How much would the power consumption of the ripple eliminator increase when the experimental design is upscaled? It should be mentioned.
Response 7 : As a scale increases, power consumption is expected to increase. It will be studied in the future research.
Point 8 : Depending on the authors' response to the previous bullet point, it would be sound to extrapolate the losses of the ripple eliminator at the power levels of the commercial products. Would the efficiency still be above 87%?
Response 8 : In this study, the circuits used in the experiment were made of PCB. Thus, the completed LED circuit is available in practice. Therefore, it is expected to produce more than 87% of products even if they are manufactured commercially.
Reviewer 2 Report
The manuscript is, without doubt, relevant. The authors propose an improved high-performance LED converter, which is characterized with a complete removal of light flicker. The circuit configuration of converter is well descripted and some analytical results are presented. Also, a 10 W prototype of the LED converter is presented. The comparison between reference converter and proposed circuit is made.
The authors show multiple results. The experimental waveforms of the output voltage, output current, and light output at 240 VAC / 50 Hz are presented. It is shown that the light flicker is completely removed. Likewise, power efficiency and efficiency difference according to the input voltage measured at 50 Hz are given. It is shown that the power efficiency of the proposed converter is 2.6 - 2.9% lower than that of the reference. Although the total power increases 2.7% on average, the power efficiency is still more than 87% throughout the range of 100-240 VAC.
The experimental results illustrate that the regulation of the LED output current is less than 0.92% at 100-240 VAC. In addition, the proposed LED converter is characterized with high power factor (>0.84) and low total harmonic distortion factor (<14.3%).
The research is in-depth and it conforms to the requirements of the IEC 61000-3-2 Class D standard at input voltages of 100 VAC and 240 VAC for high-performance LED converters.
Author Response
Point 1 : The manuscript is, without doubt, relevant. The authors propose an improved high-performance LED converter, which is characterized with a complete removal of light flicker. The circuit configuration of converter is well descripted and some analytical results are presented. Also, a 10 W prototype of the LED converter is presented. The comparison between reference converter and proposed circuit is made.
Response 1: Thank you for your posivite feedback.
Point 2 : The authors show multiple results. The experimental waveforms of the output voltage, output current, and light output at 240 VAC / 50 Hz are presented. It is shown that the light flicker is completely removed. Likewise, power efficiency and efficiency difference according to the input voltage measured at 50 Hz are given. It is shown that the power efficiency of the proposed converter is 2.6 - 2.9% lower than that of the reference. Although the total power increases 2.7% on average, the power efficiency is still more than 87% throughout the range of 100-240 VAC.
Response 2: Thank you for your comments.
Point 3 : The experimental results illustrate that the regulation of the LED output current is less than 0.92% at 100-240 VAC. In addition, the proposed LED converter is characterized with high power factor (>0.84) and low total harmonic distortion factor (<14.3%).
Response 3: Thank you again for your comments.
Point 4 : The research is in-depth and it conforms to the requirements of the IEC 61000-3-2 Class D standard at input voltages of 100 VAC and 240 VAC for high-performance LED converters.
Response 4: We appreciate your comments very much.